# How the Marketization of Land Transfer under the Constraint of Dual Goals Affects the High-Quality Development of Urban Economy: Empirical Evidence from 278 Prefecture-Level Cities in China

**Zhiqing Yan [1] and Zisheng Yang [2,*]**

1   School of Economics, Yunnan University of Finance and Economics, Kunming 650221, China
2   Institute of Targeted Poverty Alleviation and Development, Yunnan University of Finance and Economics, Kunming 650221, China
*   Correspondence: zz0976@ynufe.edu.cn; Tel.: +86-138-8896-4270

**Abstract:** As an important part of comprehensively implementing the new development concept and accelerating the construction of a new development pattern, the market-oriented allocation of land elements plays an important role in promoting high-quality economic development. This paper first constructs a theoretical model of the multi-channel impact of land transfer marketization on the high-quality development of urban economy under the framework of two departments and reveals the impact of land transfer marketization on the high-quality development of urban economy from the theoretical mechanism. Secondly, taking the panel data of China's prefecture-level cities from 2011 to 2020 as a research sample, the proportion of land bidding, auctioning, and listing area in the total area of land transfer was used to measure the marketization level of land transfer in prefecture-level cities, and the high-quality development of urban economy was measured by drawing on the regional development and people's livelihood index released by authoritative institutions, and the impact and mechanism of land transfer marketization on the high-quality development of urban economy were analyzed based on multiple measurement models and multi-dimensional empirical analysis. Finally, the paper constructs a moderating effect model to investigate the role of economic growth target constraints and environmental target constraints on the relationship between the two. The study found that: (1) There is a significant U-shaped nonlinear relationship between the marketization of land transfer and the high-quality development of urban economy. When the market-oriented development of land transfer is at a low level, it will inhibit the high-quality development of the urban economy. With the gradual improvement of the level of market-oriented development of land transfer, it will promote the high-quality development of the urban economy. (2) The dual goal constraint plays a significant regulatory role in the relationship between the marketization of land transfer and the high-quality development of the urban economy. Instead of moderate economic growth targets constraining the positive adjustment of the impact of land transfer marketization on the high-quality development of urban economies, strict environmental target constraints are more conducive to strengthening the high-quality economic development effect of land transfer marketization. (3) There are obvious heterogeneities in the impact of land transfer marketization on the high-quality development of urban economy in terms of urban location, urban scale, urban resource endowments, and the characteristics of the city itself. Based on this, it is proposed to deepen the market-oriented reform of land elements and give full play to the high-quality economic and development effect of land transfer marketization, explore the combination of scientific and reasonable economic growth goals and environmental target constraints, and give play to the positive role of dual target constraints in promoting the high-quality development of urban economy in the marketization of land transfer. According to the objective facts that the characteristics of the city itself are quite different, the market-oriented development of land transfer is scientifically released according to local conditions and brings about high-quality development effects.

**Keywords:** marketization of land transfer; high-quality economic development; economic growth target constraints; environmental objective constraints; space overflow

## 1. Introduction

Since the reform and opening up, China's economic development has made remarkable achievements, with per capita GDP increasing from 156 USD in 1978 to 10,500 USD in 2020, entering the ranks of medium-developed countries and laying a solid foundation for the high-quality development of the urban economy [1–3]. However, under the background of China's entry into a new era of development, the main contradictions in society have undergone profound changes, unbalanced and insufficient development has become the first of the contradictions, and high-quality development is facing new challenges. Specifically, China's economic development structure transformation is urgent, the cost of production factors is rising, the demand for high-quality diversification of the consumption structure is intensifying, the people's growing demand for a better life has put forward higher requirements for the speed of industrial structure upgrading, and the region is facing problems such as the widening of the development gap and the worsening of the income gap [4–7]. At the same time, after experiencing a long period of rapid growth, the main contradictions of Chinese society and the phased characteristics of economic development have undergone fundamental changes. It is precisely based on the profound changes in China's economic development fourth environment and the fundamental change of the traditional development model accumulated by the elements of "paving the way" and "project" in the past, that the 19th National Congress of the Communist Party of China came to a major conclusion that China's economy has shifted from a high-speed growth stage to a high-quality development stage. In 2020, based on the domestic development situation and grasping the general trend of international development, the central government further proposed to "accelerate the construction of a new development pattern with the domestic cycle as the main body and the domestic and international dual cycles promoting each other", which put forward a new strategic concept for the high-quality transformation of China's economy. High-quality development has become an inevitable requirement for adapting to the changes in the main contradictions of society and realizing the sustained and stable development of China's economy.

The efficiency of resource space allocation is an Important factor affecting the high-quality development of urban economy. Among them, the misallocation of land resources is one of the important sources of inhibiting the improvement of the efficiency of urban resource spatial allocation, which is closely related to the land transfer behavior of local governments [8,9]. Since the implementation of the "bidding, auctioning and listing" land market-oriented reform system ("bidding, auctioning and listing" refers to the abbreviation of the methods of bidding, auction and listing and transfer of state-owned land use rights, and the abbreviations are used in the following paragraphs), land transfer income has become an important source of income for local government finances. China's fiscal statistics show that the scale of China's land transfer revenue is rapidly expanding, and the land transfer income in 2020 reached 8.414 trillion yuan, an increase of 15.90% year-on-year, accounting for 84.03% of local fiscal revenue, setting a record for China's highest in 33 years (Source: https://new.qq.com/omn/20210203/20210203A03AHG00.html, accessed on 11 October 2022). Undoubtedly, such a huge land transfer income has increased the financial strength of local governments and alleviated the financial pressure brought about by the reform of the tax-sharing system, but the expansion of land revenue has also caused a serious problem of regional resource space mismatch, which has a deep-seated negative effect on the upgrading of regional industrial structure and innovative development, and has become a problem that cannot be ignored in the process of promoting the high-quality development of the urban economy [10,11].

At the same time, the 19th National Congress of the Communist Party of China explicitly regarded the market-oriented allocation of factors as one of the two key points of economic structural reform. The Fourth Plenary Session of the Nineteenth Central Committee of the Communist Party of China further emphasized the need to promote the construction of a factor market system and achieve factor price market decisions, independent and orderly flows, and efficient and fair allocation. In January 2022, the

General Office of the State Council issued the "Overall Plan for the Pilot Comprehensive Reform of Market-oriented Allocation of Factors", which strengthened the urgency and necessity of the market-oriented reform of factors from the practical level. Local governments are both land suppliers and responsible parties for achieving high-quality economic development and have attached great importance to promoting high-quality economic development with the help of multi-dimensional land policies [12]. Therefore, can the continuous market-oriented reform of land transfer become a winning magic weapon to empower the high-quality development of the urban economy? If the marketization of land transfer can indeed promote the high-quality development of the urban economy, then what is the mechanism of action? Are there spatial spillover effects of regional heterogeneity? What is the enabling effect of land transfer marketization under the constraints of economic growth goals and environmental targets? The in-depth exploration of the logical mechanism and operation mechanism of the above problems under the background of the new development stage has important theoretical value and practical significance for realizing the high-quality development effect of China's land resources.

The existing research on land transfer mainly focuses on the institutional reasons for land transfer and its economic effects, and it is generally believed that the reform of the tax-sharing system is the institutional cause that induces the rise of land transfer income, and the reform of the tax-sharing system leads to a mismatch between the financial power and the power of the local government, and the two-pronged approach of fiscal pressure and investment drive intensifies the enthusiasm of local governments to obtain land finance [13,14]. Under the theoretical framework of the promotion championship, the scale of land transfer is closely related to the promotion of officials, and local officials are keen on land transfer and economic performance appraisal in order to obtain promotion, which also has a negative effect on the regional social economy in the process [15,16]. In addition, the financing platform built by local governments with land use rights as the medium has evolved into "land finance", resulting in high local government debt and further expanding financial risks [17–22]. There are also studies involving land transfer and high-quality economic development, urban innovation capabilities, industrial structure upgrading, and economic fluctuations [23–26]. Further, the academic community has reached a relatively consistent understanding of the connotation of the market-oriented policy of land transfer, and the marketization of land transfer is the act of transferring the right to use land for state-owned construction in the form of market-oriented allocation of bidding, auction and listing [27,28], Focusing on the impact of land transfer marketization on the high-quality development of urban economy, scholars have also made preliminary and beneficial explorations. Zhong and Hu (2016) examined the impact of land revenue distribution system on the utilization efficiency of urban construction land [29]. Xu (2018) based on the prefecture-level city panel data measurement model found that the marketization of land transfer has a positive effect on economic growth [30]. Jiang (2019) examined the impact of land market-oriented transfer and green total factor productivity from the perspective of industrial structure optimization, which has reference value for the high-quality development of urban economy [31].

Although the existing literature shows a lot of research on the economic welfare effect of land transfer, and a small amount of useful exploration has been carried out in the economic high-quality development effect of land transfer marketization, which provides a certain theoretical reference for explaining the impact of land transfer marketization on the high-quality development of urban economy, the above research still cannot provide reliable evidence that land transfer marketization affects the high-quality development of urban economy. On one hand, the mechanism of the marketization of land transfer on the high-quality development of the urban economy is complex and there are many uncertain factors, and the theoretical assumption of the simple linear correlation between the two cannot explore a deeper mechanism of action; it is difficult to sort out the multiple improvement paths of the high-quality development of the urban economy; and it is impossible to put forward specific and effective policies for the high-quality development effect of the urban

economy of the land transfer marketization. On the other hand, the systematic empirical test of the marketization of land transfer affecting the high-quality development of urban economy also needs to be improved. In addition, the existing research ignores the typical fact that the marketization of land transfer and the high-quality development of urban economy in the Chinese context are bound to be constrained by economic growth targets and environmental targets, which is precisely an important aspect of formulating high-quality urban economic development policies tailored to local conditions within large countries. In view of this, based on the panel data of 278 prefecture-level cities in China from 2011 to 2020, this study uses a variety of econometric models to empirically analyze the impact and mechanism of land transfer marketization on the high-quality development of urban economy under the dual objective constraint.

The possible marginal contribution of this study is reflected in the following four aspects. First is to explore the nonlinear relationship between the marketization of land transfer and the high-quality development of the urban economy, and its spatial spillover effect, which not only expands the research field of the economic welfare effect of the marketization of land transfer, but also enriches the land policy support literature for the high-quality development of the urban economy. The second is to construct a theoretical model of land transfer marketization under the framework of two departments to affect the high-quality development of urban economy through multiple channels, to reveal the impact of land transfer marketization on the high-quality development of urban economy from the theoretical mechanism, and to identify the existence and contribution of the dual mechanism of resource allocation effect and industrial structure upgrading effect by using the improved intermediary effect model. The third is to further integrate into the constraints of economic growth targets and environmental targets under the Chinese context, subdivide the constraint intensity of the two major goals, and deepen the constraint mechanism and characteristics of the market-oriented land transfer to empower the high-quality development of the urban economy. Fourth, comprehensively considering the exogenous impact of the "Broadband China" strategy and the "low-carbon pilot city" policy, as well as endogenous differences such as urban location, urban resource endowments, and urban characteristics, the heterogeneity characteristics of land transfer marketization affecting the high-quality development of urban economy are systematically analyzed.

The rest of the study is arranged as follows: the second part is a theoretical analysis and the hypothesis of the study; the third part is the study design, focusing on the research methods and data sources; the fourth part combines the static panel model and the SDM space panel model to conduct empirical analysis and systematically discusses the impact of land transfer marketization on the high-quality development of urban economy. The fifth part analyzes the regulatory effect of economic growth goals and environmental objectives on the marketization of land transfer on the high-quality development of urban economy from the perspective of dual goal constraints. The sixth part systematically analyzes the heterogeneous characteristics of land transfer marketization affecting the high-quality development of urban economy from the endogenous differences such as the Qinling-Huaihe River dividing line, resource endowments, and the characteristics of the city itself. Section 6 summarizes the full text and makes policy recommendations.

## 2. Theoretical Analysis and Research Hypothesis

### 2.1. China's Land Transfer Market-Oriented Reform

The reform of the land system and the marketization of land transfer are closely related to economic development, the land system is an important institutional force to activate economic vitality, and China's unique land system has created a model of local governments "seeking development with land". Since the reform and opening up in 1978, the development of China's land transfer market has gone through three stages: "the auxiliary role of the market", "the basic role of the market", and "the decisive role of the market".

The period of socialist revolution and construction in China (1949–1977): From private ownership of land to planned control. Urban land use is managed by lease, and users are required to pay rent. In 1954, with the establishment of socialist public ownership, the state nationalized urban land through confiscation, redemption, and other forms, and implemented a highly centralized planning management model. By combining urban land management with the planned economy, urban land allocation has obvious characteristics of "three noes" (gratuitous, indefinite, and non-flow), which has continued until the reform of the urban land use system after the reform and opening up.

China's Reform Exploration and Development Period (1978–2011): Move from planned to marketable configuration. In 1981, the practice of paid use of state-owned land was carried out in Shenzhen, Hefei, Fushun, Guangzhou, and other places. The paid use of land breaks the original administrative allocation method and provides some experience for the reform of the land system, but does not involve the land market mechanism. In 1982, the 12th National Congress of the Communist Party of China clearly pointed out that the relationship between planning and the market is planned first, the market is supplemented, and the role of the market cannot be ignored, which means that China's economic system has gradually shifted from a "planned economy" to a socialist market economy with "planning as the mainstay and the market as a supplement". With the progress of reform, the drawbacks of the administrative allocation method in the process of practice have gradually been revealed, and problems such as inefficient land use and serious waste of land resources have emerged, the reform of the administrative allocation of land use method is imperative, and the promotion of the contracting system in rural areas has provided a useful reference for the reform of state-owned construction land. In 1987, Shenzhen City, Guangdong Province, completed the first transaction of the 50-year use right of state-owned land by market means, and for the first time, allocated resources by market means, and the obstacle prohibiting the transfer of land use rights under the current law was broken. With the development of economic structural reform and the rapid development of urbanization in China, land leasing, shareholding and other methods have begun to become active. The Land Administration Law, amended in 1988, clearly stipulates the implementation of the system of paid use of state-owned land, which provides a legal guarantee for the development of the land market. After entering the 21st century, the state-owned land market developed rapidly, and from 2005 to 2013, the scale, price, and marketization of the land market increased by 113.66%, 274.09%, and 37.07%, respectively. The mechanism of "land acquisition–reserve–development–transfer" has gradually taken shape, and the transfer of urban land "bidding, auctioning, and listing" has gradually become active.

A new period of China's comprehensive deepening of reform (2012–present): from the separation of urban and rural areas to the integration of urban and rural development. In 2012, the 18th National Congress of the Communist Party of China proposed to "promote the integrated development of urban and rural areas", and while developing cities, it is also necessary to take into account rural areas. In 2013, the proposal of "establishing a unified urban and rural construction land market" kicked off the prelude to comprehensively deepening reform, and the market allocation of resources gradually shifted from "basic" to "decisive". In 2015, the central government selected 33 regions across the country as pilot projects for the rural "three plots" reform. Since the pilot project was launched, the first auction of the right to use rural collective operation construction land was successfully traded in Meitan County, Guizhou Province. In the same year, the central government selected 15 regions across the country to carry out the pilot project of "homestead system reform", and through exploration, each region increased the property income of farmers to a certain extent. As one of the pilot areas in Yiwu, Zhejiang, the design scheme of the "separation of three rights" system for residential land provides a reference for the reform of other regions and provides certain experience for other regions to carry out work. Moreover, the obstacle that collective construction land cannot have the same rights and the same price as state-owned construction land was broken down in the Land Administration

Law amended in 2019, which also provided a guarantee for the integrated development of urban and rural areas.

### 2.2. The Marketization of Land Transfer and the High-Quality Development of the Urban Economy

After the reform of fiscal decentralization and tax sharing system, local governments generally have the behavior and decision-making of transferring industrial land at low prices and transferring land at high prices, which has profoundly affected the upgrading of regional industrial structure [32–34]. It can be inferred from this that the local government agreement to transfer land lacks competitiveness, cannot fully reflect the market value of land, reduces the efficiency of land resource allocation, and allows market orientation to rely on the price mechanism to screen land buyers, which can restrain the transferee to improve production efficiency, expand the driving role of high value-added industrial enterprises, block the easy entry of low value-added enterprises, and realize the upgrading of regional industrial structure as a whole and promote the high-quality development of urban economy. Further, this study distinguishes the degree of industrial structure optimization, that is, the rationalization and high-level of industrial structure, both of which can reflect the improvement of production technology and industrial layout adjustment, but the former is manifested as the orderly flow and effective allocation of production factors between different departments to achieve a benign interactive state, and the latter is the reconfiguration process of production factors to a higher gradient industry, reflecting the social-led industrial upgrading [35–37]. Since the high-quality development of urban economy emphasizes the coordinated acquisition of multiple systems such as economic benefits, social benefits, and environmental benefits, on the one hand, it requires the scientific flow of resource element allocation to achieve economic benefits, on the other hand, it needs corresponding environment-friendly industrial transfer and restructuring to reduce the environmental pressure in the process of obtaining economic benefits. Therefore, the analysis of the impact of land transfer marketization on the high-quality development of urban economy should start from whether the marketization of land transfer will promote the transfer of production factors to environment-friendly enterprises. Relative to enterprises, in the short term, the marketization of land transfer objectively promotes the rise of land prices, to a certain extent, and forces traditional enterprises to pay more attention to the level of corporate profitability within the limits of environmental penalties, ignoring environmental protection. Therefore, with the concentration of production factors to the high economic efficiency of the sector, environment-friendly enterprises face the dilemma of insufficient production factors. From this level, land transfer marketization level is low and will lead to the rationalization of industrial structure reverse development, inhibiting the high-quality development of urban economy. However, in the long run, when the level of land transfer marketization is high, the land price naturally rises, and the land price cost constraint formed will force traditional enterprises to carry out technological innovation and industrial structure optimization, and gradually transform to a low-carbon economy, such as labor-intensive industries to capital-intensive industries, and will also eliminate some traditional enterprises that cannot invest in research and development [38]. At this time, the process of industrial structure can reduce carbon emissions in economic growth, and the positive spillover effect of technological iterative renewal has promoted the high-quality development of urban economy. Based on this, Hypothesis 1 of the study is proposed:

**Hypothesis 1**. *The marketization of land transfer has a non-linear relationship with the high-quality development of urban economy.*

### 2.3. The Marketization of Land Transfer Affects the Transmission Mechanism of High-Quality Urban Economic Development

This study further reveals the impact of land transfer marketization on the high-quality development of urban economy by constructing a theoretical model of multi-channel

influence on the high-quality development of urban economy. Build a production function containing the energy sector on top of the Douglas production function:

$$Y = TL^{\alpha}K^{\beta}E^{1-\alpha-\beta} \tag{1}$$

$$CP = \frac{Y}{E^{1-\alpha-\beta}v} = \frac{TL^{\alpha}K^{\beta}}{v} \tag{2}$$

$$T = f(\vartheta) \tag{3}$$

Among them, *Y, T, L, K, E, CP, v* represent economic output, land use efficiency, labor force, capital stock, high-quality economic development, and economic high-quality development conversion coefficients, respectively. It is assumed that land use efficiency is positively correlated with the marketization of land transfer ($\vartheta$). Simplifying the above equation yields:

$$CP = f(\vartheta)\frac{L^{\alpha}K^{\beta}}{v} \tag{4}$$

From Equation (4), it can be seen that the greater the marketization of land transfer ($\vartheta$), the higher the land use efficiency (T), and the higher the level of high-quality economic development of a single sector.

Extend the model again to the two-sector model and distinguish between the high and low sectors of high-quality economic development, that is $CP_1 > CP_2$, the following:

$$\frac{CP_1}{CP_2} = \lambda > 1 \tag{5}$$

Combined with the above theoretical analysis, it can be seen that the marketization of regional land transfer will affect the upgrading of industrial structure, and in the model push, there are:

$$UIS = \frac{Y_1}{Y_2} = g(\vartheta). \tag{6}$$

$$CP = \frac{Y}{vE} = \frac{Y_1 + Y_2}{v(Y_1/CP_1v + Y_2/CP_2v)} \tag{7}$$

Integrate (5)–(7) to get:

$$\begin{aligned} CP &= \frac{Y}{vE} = \frac{Y_1+Y_2}{v(Y_1/CP_1v + Y_2/CP_2v)} = \frac{UIS+1}{Y_1/CP_1 + Y_2/CP_2} = \left(\frac{UIS+1}{UIS+\lambda}\right)CP_1 \\ &= \left[\frac{g(\vartheta)+1}{g(\vartheta)+\lambda}\right]f(\vartheta)\frac{L_1^{\alpha}K_1^{\beta}}{v}. \end{aligned} \tag{8}$$

Since $\lambda > 1$, it can be seen that the higher the marketization of land transfer ($\vartheta$), the better the industrial structure optimization and upgrading (UIS), and the better the high-quality economic development, indicating that the higher the proportion of departments with high economic high-quality development, the higher the overall economic development of the region.

In summary, Hypothesis 2 of the study is proposed:

**Hypothesis 2.** *The marketization of land transfer mainly affects the high-quality development of urban economy through the effect of resource allocation and industrial structure upgrading.*

### 2.4. Under the Constraint of Dual Goals, the Marketization of Land Transfer Affects the Differentiation Mechanism of High-Quality Urban Economic Development

High-quality economic development not only needs to give play to the basic role of the market in allocating resources, but also cannot be separated from the strong promotion of the government [39,40]. The formulation of economic growth targets and environmental targets is an important policy tool for the government to promote high-quality development, and appropriate dual target constraints play an important role in achieving the goal

of high-quality economic development [41–47]. Under the promotion competition theory, local governments have the initiative to vigorously complete the economic growth target, when the central government issues the target values of the provinces, the local governments will often increase the number of layers, exceed the targets, and gradually form a development model of "competition for growth" and "competition for investment", which has created China's rapid economic growth in a certain historical period, and inevitably led to the emergence of a "zero-sum game", resulting in the distortion of the investment structure of "heavy infrastructure and light service", and inhibiting the improvement of total factor productivity. At the same time, long-term traditional factor input, economic growth has formed a path dependence of "high energy consumption, high emissions, and high pollution", which further leads to environmental pollution problems. Therefore, the excessively high economic growth target constraints weaken the resource allocation effect and industrial structure upgrading effect brought about by the marketization of land transfer and inhibit the high-quality development of the urban economy [48]. In contrast, environmental target constraints have improved the quality of the ecological environment through the combination of environmental regulation policies, environmental protection inspection systems, and environmental governance efforts, but energy consumption and environmental pollution issues cannot be ignored, and the "14th Five-Year Plan" clarifies the mandatory environmental target constraints on China's energy consumption and carbon emissions. In general, environmental target constraints will force enterprises to make pollutant discharge adjustments and green production, and will also force enterprises to increase investment in green technology innovation and research and development, and accelerate the advanced development of industrial structure [49]. Based on this, the study Hypothesis 3 is proposed:

**Hypothesis 3.** *The excessively high economic growth target constraints weaken the efficiency of the high-quality development of the urban economy marketed by land transfer, and the strict environmental target constraints strengthen the high-quality development effect of the urban economy market-oriented land transfer.*

## 3. Research Design
### 3.1. Variable Design
3.1.1. Static Panel Model

This study is based on the STIRPAT model, which comprehensively considers the impact of major socio-economic factors on the environment [50], and can be used to study the high-quality development of urban economy. The benchmark model is set as follows:

$$\ln HED_{it} = \alpha_0 + \alpha_1 \ln LT_{it} + \alpha_2 \ln SLT_{it} + \alpha_3 \ln C_{it} + \mu_i + \lambda_t + \delta_{it} \tag{9}$$

Among them, $HED_{it}$ represents the level of high-quality economic development of the city i in the t year; and respectively, $LT_{it}$ and $SLT_{it}$ are the marketization of land transfer and its square items; $C_{it}$ represents the control variable group that affects the high-quality development of the urban economy, including variables such as the level of economic development, the level of urbanization, the size of the population, fixed asset investment, and outward direct investment; $\mu_i$, $\lambda_t$ and $\delta_{it}$ represent the regional effect, time effect, and random perturbation term.

3.1.2. Mechanism to Test the Model

In order to test the existence and contribution of the two major mechanisms of resource allocation effect and industrial structure upgrading effect, this study draws on existing research methods [51]. Add the mechanism variable $\gamma_{it}$ to Equation (9). The mechanism variables include the resource allocation effect of total factor productivity ($RAE$) measured by the Soro residual method,the ratio of added value of the secondary and tertiary industries reflects the effect of industrial structure upgrading ($ISU$), other variables remain unchanged,

and the test steps are: ① Directly test the impact of land transfer marketization on the effect of resource allocation and industrial structure upgrading, so as to verify the existence of the two major mechanisms; ② on the basis of the regression Equation (11), the explanatory force of the two major mechanisms is further measured, and the calculation procedure is to obtain the coefficient of land transfer marketization $\hat{\alpha}$ and $\hat{\varphi}$ from the regression Equations (9) and (11), and calculate $1 - \hat{\alpha}/\hat{\varphi}$, the contribution of the mechanism variable can be obtained, and the detailed proof process can be found in the appendix section of the document by Cutler and Liters-Muney. The relevant regression equations are as follows:

$$\ln \gamma_{it} = \beta_0 + \beta_1 \ln LT_{it} + \mu_i + \lambda_t + \delta_{it} \tag{10}$$

$$\ln HED_{it} = \varphi_0 + \varphi_1 \ln LT_{it} + \varphi_2 \ln SLT_{it} + \varphi_3 \ln \gamma_{it} + \varphi_4 \ln C_{it} + \mu_i + \lambda_t + \delta_{it} \tag{11}$$

### 3.1.3. Space Econometric Model

The SDM model is a cutting-edge model in the current development of space measurement, which not only combines the advantages of SAR and SEM spatial models, but also properly handles the related shortcomings, such as considering the spatial impact of random shocks [52]. Therefore, on the basis of the SDM model, this paper constructs a spatial econometric model of land transfer marketization affecting the high-quality development of urban economy, and the equation is as follows:

$$
\begin{aligned}
\ln HED_{it} = &\ \alpha_0 + \rho W \ln HED_{it} + \alpha_1 \ln LT_{it} + \alpha_2 \ln SLT_{it} + \alpha_3 \ln C_{it} \\
&+ \varepsilon_1 W \ln LT_{it} + \varepsilon_2 W \ln SLT_{it} + \varepsilon_3 W \ln C_{it} + \mu_i + \lambda_t + \delta_{it}
\end{aligned}
\tag{12}
$$

Among them, all variables are consistent with (9), $\rho$ is spatial autoregression coefficients and $W$ is spatial weight matrix. This study mainly uses two spatial weight representation methods, one is the geographical distance weight matrix, whether the city is adjacent as the standard, the corresponding assignment is 1 and 0, mainly used in the benchmark regression of the SDM space model, and the second is the economic distance spatial weight matrix, used in the robustness test: the reciprocal of the per capita GDP gap between the two cities is constructed as the standard.

### *3.2. Variable Design*

3.2.1. Explained Variable

The explained variable of this study is the high-quality development of urban economy (*HED*). Since the 19th National Congress, high-quality development has become the direction of China's future development. The existing discussions on the connotation of high-quality development mainly have three types of perspectives: the first type combines the "five major concepts" with the main contradictions of society. The second category equates high-quality economic development with high-quality development content. The third category analyzes high-quality development from the macro, meso, and micro levels. In fact, the overall meaning of the three classification perspectives is consistent, and its essential connotation is high efficiency, fairness, and green sustainable development aimed at meeting the people's growing needs for a better life. The Regional Development and People's Livelihood Index jointly released by the Chinese Statistical Society and the National Bureau of Statistics, the evaluation index system covers economic development, people's livelihood improvement, social development, ecological construction, scientific and technological innovation, and public evaluation of 6 aspects, involving 42 indicators, as the agent variable of high-quality urban economic development, not only coincides with the essential connotation of high-quality development, but also reached a certain consensus in the academic community. Therefore, this paper draws on the regional development and people's livelihood index to measure the high-quality development of the urban economy and expands the index to 2020.

### 3.2.2. Explanatory Variable

The explanatory variable in this study is the marketization of land transfers (*LT*), There are many measurement methods related to the marketization of land transfer, for example, Qian and Mou (2012) adopted the price weighting method to take the auction price as the normal price, and the other forms of transaction price and the normal price were determined by the weighting to measure the marketization of land transfer [53]. Li (2017) measured the marketization of land transfer by using the income from land transfer and the number of land transfers [54], and Huang (2020) measured the marketization of land transfer by using the proportion of land market-oriented transaction area to the total area [55]. Noting the advantages and disadvantages of the above practices, while considering that the proportion of land parcels cannot fully reflect the size of the parcel area, and the level of land transfer marketization is closely related to the parcel area, and the land price weight law will deviate due to the economic development gap between cities, this paper uses the proportion of land bidding, auction, and listing transfer area to the total transfer area to measure the land transfer marketization.

### 3.2.3. Controlled Variable

By combing the literature on the influencing factors of high-quality urban economic development, it is found that the high-quality development of urban economy is mainly affected by economic development, population size, and technological progress, so this study selects relevant control variables on this basis (as shown in Table 1).

**Table 1.** Description of control variables.

| Variables | Code | Illustrate |
|---|---|---|
| Level of Economic Development | RGDP | GDP per capita |
| Level of Urbanization | UR | Urbanization rate |
| Population Size | POP | Urban permanent population |
| Investment in Fixed Assets | IFA | The amount of fixed asset investment per capita |
| Foreign Direct Investment | FDI | Outward FDI as a share of GDP |

### 3.3. Data Sources

On one hand, this study considers the characteristics of the rapid development of land transfer marketization in recent years, and on the other hand, takes into account the availability of data. Therefore, 2011–2020 was selected as the research period, the prefecture-level cities with serious lack of indicators were deleted, and 278 cities above the prefecture level in China were finally selected as the research sample. Among them, the main data were from the 2012–2021 "China Urban Statistical Yearbook", "China Land and Resources Statistics Yearbook", "China Statistical Yearbook", local municipal statistical annual reports, green patent databases, WIND databases, and EPS databases, involving the use of neighboring mean method and linear interpolation method to complete some missing values. Economic growth targets and environmental targets come from the work reports of local municipal governments. The descriptive statistics of the variables are shown in Table 2.

**Table 2.** Descriptive statistics for variables.

| Variables | Code | Obs | Mean | SD | Min | Max |
|---|---|---|---|---|---|---|
| High-quality Development of Urban Economy | lnHED | 2780 | 0.4133 | 0.2603 | 0.2736 | 0.8622 |
| Marketization of Land Transfer | lnLT | 2780 | 0.1436 | 0.0718 | 0.0237 | 0.6732 |
| The Market-oriented Square of Land Transfer | lnSLT | 2780 | 0.0267 | 0.0388 | 0.0011 | 0.4281 |
| Resource Allocation Effects | lnRAE | 2780 | 0.5366 | 0.6022 | 0.0061 | 0.8815 |
| The Effect of Industrial Structure Upgrading | lnISU | 2780 | 1.7755 | 1.3181 | 0.7276 | 3.8896 |
| Tevel of Economic Development | lnRGDP | 2780 | 4.0166 | 2.6683 | 3.3477 | 9.4599 |
| Level of Urbanization | lnUR | 2780 | 0.6612 | 0.4391 | 0.1122 | 1.7823 |
| Population Size | lnPOP | 2780 | 4.8754 | 1.0967 | 2.2971 | 7.6918 |
| Investment in Fixed Assets | lnIFA | 2780 | 3.0266 | 1.7643 | 2.3115 | 6.6571 |
| Foreign Direct Investment | lnFDI | 2780 | 1.7933 | 1.6542 | 0.2611 | 5.0072 |

## 4. Empirical Results

*4.1. The Impact of Land Transfer Marketization on the High-Quality Development of Urban Economy under the Static Panel Model*

### 4.1.1. Baseline Regression Analysis

Columns (1) and (2) of Table 3 are the regression results of the addition of the marketization of land transfer and its square terms, respectively. The regression results in column (1) of Table 3 show that the regression coefficient of land transfer marketization to the high-quality development of urban economy is 0.0266, which fails the significance test, indicating that there is a certain positive impact of land transfer marketization on the high-quality development of urban economy, but the statistical model does not support this relationship. Column (2) of Table 3 is the return result of adding the marketization square term of land transfer, the results show that the primary item of land transfer marketization is significantly positive at the 1% level, and the square term of land transfer marketization is significantly negative at the 5% level, and it is still true after controlling other relevant factors affecting the high-quality development of the urban economy, indicating that there is a significant U-shaped nonlinear relationship between the marketization of land transfer and the high-quality development of the urban economy, and Hypothesis 1 is verified. The possible reason is that when the level of marketization of land transfer is low, the factors of production are concentrated and transferred to sectors with high economic benefits, and environment-friendly enterprises are facing the dilemma of insufficient production factors, resulting in the rationalization of industrial structure and reverse development, inhibiting the high-quality development of urban economy. When the level of land transfer marketization is high, the resulting land price cost constraint will force traditional enterprises to carry out technological innovation and industrial structure optimization, and gradually transform to a low-carbon economy, the process of industrial structure upgrading can improve total factor productivity, and the positive spillover effect of technological iterative renewal has promoted the high-quality development of urban economy.

**Table 3.** Static panel model regression results.

| Variables | (1) | (2) | Tool Variable Regression | |
| :---: | :---: | :---: | :---: | :---: |
| | | | **Phase I** | **Phase II** |
| lnLT | 0.0266 | −0.6233 ** | / | −0.5977 *** |
| | (0.2135) | (0.2011) | | (0.1833) |
| lnSLT | / | 1.2044 *** | / | 1.1182 *** |
| | | (0.3316) | | (0.1721) |
| Control Variables | control | control | control | control |
| Constant Term | 1.6616 *** | 1.6861 *** | 0.0611 ** | 0.1066 ** |
| | (0.3321) | (0.4211) | (0.0672) | (0.1001) |
| Tool Variables | / | / | 0.0053 *** | / |
| | | | (0.0068) | |
| Year FE | YES | YES | YES | YES |
| City FE | YES | YES | YES | YES |
| $R^2$ | 0.5088 | 0.4726 | 0.6433 | 0.6131 |
| N | 2780 | 2780 | 2780 | 2780 |

Note: the brackets in the estimation results are the robust standard errors of individual clustering; **, ***, indicating a rejection of the original hypothesis at the significance levels of 5%, and 1%, respectively.

### 4.1.2. Tool Variable Regression

The marketization of land transfer and the high-quality development of urban economy will be affected by factors such as urban institutional environment, government governance capacity and scientific and technological innovation, which in turn will lead to endogenous problems between the marketization of land transfer and the high-quality development of urban economy. In order to alleviate the possible endogenous impact, this study draws on the practice of Zhang and Yu (2019) [56], using the mean of urban land

slope as the tool variable of the supply side of local government land transfer, and the interaction between urban land slope and economic growth target as the tool variable of the demand side of land transfer income, and using the two-stage least squares method (2SLS) for regression. Reasons for selection: On the one hand, the land slope is derived from natural factors, and there is no direct correlation between other economic variables in theory, so it satisfies the exogenous principle; on the other hand, the terrain undulation reflects the complexity of the urban terrain, which will affect the difficulty and cost of the market-oriented development of land transfer, so the correlation principle is satisfied. The regression results are shown in Table 3, and the first-stage regression results show that there is a significant positive correlation between the instrumental variable and the endogenous variable land transfer marketization, which is in line with the correlation hypothesis. The second stage regression results show that the primary term coefficient of land transfer marketization is significantly negative at the 1% level, and the square term is significantly positive at the 1% level, which is similar to the benchmark regression results, except that the regression coefficients have increased, indicating that after the use of instrumental variables to alleviate endogenous problems, there is still a stable U-shaped nonlinear relationship between land transfer marketization and high-quality urban economic development. At the same time, they have passed the tests of weak tool variables and exogenous properties, indicating that the selection of tool variables is scientific.

### 4.2. The Impact of Land Transfer Marketization on the High-Quality Development of Urban Economy under the Space Panel Model

#### 4.2.1. Spatial Correlation Analysis

Before performing spatial metrological regression, it is necessary to perform a spatial correlation test on the core variables, and the Moran' I (Moran's I exponential formula: $\text{Moran's I} = \frac{\sum_{i=1}^{278} \sum_{j=1}^{278} W_{ij}(Y_i - \overline{Y})(Y_j - \overline{Y})}{S^2 \sum_i^{278} \sum_j^{278} W_{ij}}$). exponential method is mainly used to investigate the spatial correlation characteristics between the core variables. In this study, the global Moran' I index values of 254 cities above the prefecture level in China from 2011 to 2020 were calculated by combining the geographical distance matrix. The test results show that during the sample observation period, most of the Moran' I index values of the core variables are significantly positive, indicating that there is a non-negligible spatial correlation between the marketization of land transfer between cities in China and the high-quality development of the economy, which further proves that the spatial measurement method is appropriate.

#### 4.2.2. Spatial Effects Test Results

Table 4 is the spatial measurement regression result based on the SDM model. Column (1) shows that the primary term coefficient of land transfer marketization is significantly negative at the 1% level, and its square term is significantly positive at the 1% level, and there is a significant U-shaped nonlinear relationship between the marketization of land transfer and the high-quality development of the urban economy, which is consistent with the previous empirical results. From column (2), it can be seen that the regression coefficient of the spatial lag term of the land transfer marketization has a U-shaped nonlinear relationship, but it is not significant, indicating that during the study period, the spatial spillover effect of the marketization of urban land transfer in China on the high-quality economic development of surrounding cities is not obvious. Column (3) to (5) is listed as the result of the spatial effect decomposition of the impact of land transfer marketization on the high-quality development of the urban economy, and the results show that the direct effect and total effect of the land transfer marketization of the primary item and the square term are significant at the 5% level, indicating that the relationship between the land transfer marketization and the high-quality development of the urban economy is a U-shaped nonlinear relationship in the region and in general, while in terms of indirect effects, only the square term of the land transfer marketization is significant at the 10% level. It is further explained that the spatial spillover effect of land transfer

marketization affecting the high-quality development of urban economy is small, and the possible explanation is that the siphon effect caused by the difference in the level of urban economic development is more obvious, which weakens the positive spillover impact of the market-oriented economic high-quality development of central urban land transfer.

**Table 4.** Spatial SDM model regression results.

| Variables | (1) High-Quality Development of Urban Economy | (2) Spatial Lag Items | (3) Direct Effects | (4) Indirect Effects | (5) Total Effect |
|---|---|---|---|---|---|
| lnLT | −0.5089 *** (0.2926) | −0.4926 (0.3121) | −0.5501 ** (0.3441) | −0.4216 (0.3207) | −0.97177 ** (0.5652) |
| lnSLT | 1.1668 *** (0.0167) | 1.1774 (0.0263) | 1.2613 ** (0.0499) | 1.1763 * (0.0290) | 2.4376 ** (0.1922) |
| Control Variables | control | control | control | control | control |
| rho | 0.1201 *** (0.0158) | / | / | / | / |
| Sigma2_e | 0.0011 *** (0.0011) | / | / | / | / |
| $R^2$ | 0.3387 | 0.3761 | 0.3217 | 0.3697 | 0.3811 |
| N | 2780 | 2780 | 2780 | 2780 | 2780 |

Note: the brackets in the estimation results are the robust standard errors of individual clustering; *, **, ***, indicating a rejection of the original hypothesis at the significance levels of 10%, 5%, and 1%, respectively.

### 4.3. Robustness Test

In order to further test the soundness of the impact of land transfer marketization on the high-quality development of urban economy, this study carried out a robustness test from four aspects: replacing the interpreted variables, adding the interaction between the temporal trend term and the control variable, excluding other policy influences, and replacing the spatial weight matrix.

#### 4.3.1. Replace the Interpreted Variable

Drawing on the measurement method of Shangguan and Ge (2020) [57], environmental factors were incorporated into the traditional TFP framework to measure green total factor productivity as an alternative variable for high-quality urban economic development, and labor, capital, and energy are selected in terms of input factors, and GDP and "three wastes" emissions are selected for expected output and non-expected output, respectively, and the SBM-DEA method is used for measurement. Column (1) of Table 5 shows that after replacing the interpreted variables, the marketization of land transfer still has a significant U-shaped nonlinear relationship for the high-quality development of urban economy, which effectively supports the benchmark regression results.

**Table 5.** Robustness test regression results.

| Variables | (1) Replace the Interpreted Variable | (2) Control Variables * Time Trend Item | (3) Exclude Other Policy Implications | | (4) Economic Distance Matrix Regression | | |
|---|---|---|---|---|---|---|---|
| | | | Broadband China | Low Carbon Pilot | Direct Effects | Indirect Effects | Total Effect |
| lnLT | −0.7122 ** (0.3276) | −0.7533 ** (0.3595) | −0.7015 ** (0.3173) | −0.7035 ** (0.3181) | −0.6642 *** (0.3324) | −0.6077 (0.3127) | −1.2719 *** (0.5206) |
| lnSLT | 1.2472 *** (0.4963) | 1.2668 *** (0.5017) | 1.3064 *** (0.4713) | 1.3162 *** (0.4788) | 1.2303 *** (0.0573) | 1.2184 * (0.0366) | 2.4487 *** (0.1912) |
| T * Control Variables | / | YES | / | / | / | / | / |
| Control Variables | control | control | control | control | control | control | control |
| $R^2$ | 0.3376 | 0.3388 | 0.3613 | 0.3702 | 0.3361 | 0.3046 | 0.3368 |
| N | 2780 | 2780 | 2780 | 2780 | 2780 | 2780 | 2780 |

Note: the brackets in the estimation results are the robust standard errors of individual clustering; *, **, ***, indicating a rejection of the original hypothesis at the significance levels of 10%, 5%, and 1%, respectively.

### 4.3.2. Added Control Variables to Interact with Time Trends

Studies have found that adding the interaction term between time trend and control variable in the empirical model can effectively reduce the estimation bias, because the temporal trend of the influencing factors of the interpreted variable can be relatively fixed after addition [58]. This study draws on this method for regression, and column (2) of Table 5 shows that the significance and direction of the coefficient of the impact of land transfer marketization on the high-quality development of urban economy have not changed substantially, indicating the robustness of the benchmark regression results.

### 4.3.3. Exclude Other Policy Implications

Other policies mainly considered in this study include "Broadband China" and "Low Carbon Pilot City", because these two policies have a greater impact on the marketization of land transfer and the low-carbon transformation of cities, and need to be taken into account. The "Broadband China" strategy was promoted in batches with pilot cities from 2014 to 2016, with a value of 1 for the pilot cities and 0 for the other day. The "low-carbon pilot cities" began in 2010 by selecting 5 provinces and 8 cities to carry out pilot work, followed by the implementation of the second and third batches of pilot cities in 2012 and 2017, respectively, and the corresponding inclusion of pilot cities with a value of 1 and vice versa of 0. Column (3) of Table 5 is the return result excluding the impact of other policies, and the results show that there is no significant difference in the regression coefficient of the impact of land transfer marketization on the high-quality development of the urban economy after excluding the two important related policies, which proves that the above return results are stable.

### 4.3.4. Replaces the Spatial Weights Matrix

Different spatial weight matrix choices may affect spatial regression results, and this study re-performs SDM regression by replacing the geographic distance matrix with an economic geographic matrix. The regression results in column (4) of Table 5 show that the regression coefficients and directions of the primary and square terms of the marketization of land transfer are basically the same as those of the previous text, and the significance is enhanced, which further proves the robustness of the benchmark regression results.

### 4.4. Mechanism of Action Analysis

The empirical results mentioned above show that there is a steady U-shaped nonlinear relationship between the marketization of land transfer and the high-quality development of urban economy. The marketization of land transfer can affect the high-quality development of urban economy through two channels: resource allocation effect and industrial structure upgrading effect. Therefore, this study identifies the existence and contribution of the two major mechanisms of action on the basis of the previous mechanism test model. Table 6 is the result of the test of the existence and contribution of the mechanism of the impact of land transfer marketization on the high-quality development of the urban economy. Columns (1) and (2) are the regression results of testing the existence of the mechanism, and the results show that the marketization of land transfer has a significant positive impact on the variables of the two major mechanisms, and the impact on the effect of resource allocation is greater than the effect of industrial structure upgrading, indicating that the marketization of land transfer has a positive impact on the mechanism. Columns (3)–(5) are the regression results of the contribution of the measurement mechanism, and the results show that the contribution of the two major mechanisms to the marketization of land transfer to affect the high-quality development of the urban economy exceeds 50%, of which the contribution of the resource allocation effect is the largest, reaching 38.2166%, followed by the industrial structure upgrading effect of 20.5533%. At this point, Hypothesis 2 is verified.

**Table 6.** Regression results of the existence and contribution of the mechanism of action.

| Explanatory Variable | (1) lnRAE | (2) lnISU | (3) Baseline Regression | (4) Mechanism 1: Resource Allocation Effect | (5) Mechanism 2: The Effect of Industrial Structure Upgrading |
|---|---|---|---|---|---|
| | | | | **lnRAE** | **lnISU** |
| lnLT | 0.0573 *** (0.0367) | 0.0088 *** (0.0031) | −0.6233 ** (0.2011) | −1.0088 *** (0.3921) | −0.7846 ** (0.3829) |
| lnSLT | / | / | 1.2044 *** (0.3316) | 1.9494 *** (0.5633) | 1.5160 ** (0.5201) |
| lnRAE | / | / | / | 0.0533 *** (0.0397) | / |
| lnISU | / | / | / | / | 0.0076 *** (0.0027) |
| $1 - \hat{\alpha}/\hat{\varphi}$ | / | / | / | 38.2166% | 20.5533% |
| Control Variables | control | control | control | control | control |
| Year FE | YES | YES | YES | YES | YES |
| City FE | YES | YES | YES | YES | YES |
| N | 2780 | 2780 | 2780 | 2780 | 2780 |
| Adj $R^2$ | 0.7131 | 0.7362 | 0.4926 | 0.5077 | 0.4962 |

Note: the brackets in the estimation results are the robust standard errors of individual clustering; **, ***, indicating a rejection of the original hypothesis at the significance levels of 5%, and 1%, respectively.

### 4.5. Heterogeneity Analysis

4.5.1. Heterogeneity of Urban Location

The Qinling-Huaihe line is an important geographical dividing line between the north and south of China, and it is also a natural dividing line for whether central heating is provided in winter, and the terrain, climate, economic development, and ecological environment on both sides of the Qinling-Huaihe River line have unique differences. Whether there is a difference in the high-quality economic development effect of land transfer marketization on both sides of the dividing line is worth exploring in depth. Column (1) of Table 7 is the result of the return of urban location heterogeneity, which shows that compared with the south of the Qinling-Huaihe line, the inflection point value of the land transfer marketization north of the Qinling-Huaihe line affects the high-quality development of the urban economy is even greater, and the inflection point of the impact of land transfer marketization on the high-quality development of the urban economy is wider in advance, indicating that the economic high-quality development effect of the land transfer marketization in northern cities is higher than that of southern cities. The possible reason is that the industrial structure of northern cities is relatively low-end, mainly in the secondary industry, facing serious environmental pollution problems, so there is a strong demand for pollution control and emission reduction.

**Table 7.** Heterogeneity test regression results I.

| Variables | (1) Heterogeneity of Urban Location | | (2) Heterogeneity of Urban Scale | |
|---|---|---|---|---|
| | South of the Qinling-Huai River Line | North of the Qinling-Huai River Line | Metropolis | Small and Medium-Sized Cities |
| lnLT | −0.5265 * (0.0375) | −0.4036 ** (0.1789) | −0.4123 *** (0.2153) | −0.5017 ** (0.2893) |
| lnSLT | 1.0066 * (0.3167) | 1.0012 ** (0.2018) | 1.0037 *** (0.2169) | 1.0058 ** (0.2789) |
| Control Variables | control | control | control | control |
| $R^2$ | 0.4138 | 0.3989 | 0.3867 | 0.4312 |
| N | 2780 | 2780 | 2780 | 2780 |

Note: the brackets in the estimation results are the robust standard errors of individual clustering; *, **, ***, indicating a rejection of the original hypothesis at the significance levels of 10%, 5%, and 1%, respectively.

### 4.5.2. Heterogeneity of Urban Scale

The scale of the city itself will lead to different economic and high-quality economic development effects of land transfer marketization. Compared with small- and medium-sized cities, large cities have scale advantages in industrial structure, environmental governance investment, and scientific and technological innovation, forming a certain agglomeration effect, which is conducive to the optimal allocation of resources, and at the same time, huge energy consumption and land supply demand also lead to congestion effects, resulting in aggravation of urban environmental problems. Is the high-quality economic development effect of land transfer market-oriented affected by the urban agglomeration effect and the congestion effect? Which is more important than the other? It is a meaningful exploration. Based on the 2014 Notice of the State Council on Adjusting the Criteria for Grading Urban Size, this paper classifies urban permanent residents with a population of less than 1 million into small- and medium-sized cities, and those with a permanent population greater than 1 million into the category of large cities. Column (2) of Table 7 is the result of the return of urban scale heterogeneity, which shows that compared with small- and medium-sized cities, the economic high-quality development effect of land transfer marketization in large cities is greater, which shows that the rapid development of land transfer marketization can promote the advanced development of urban industrial structure, which can effectively obtain agglomeration effects and alleviate the impact of congestion effects.

### 4.5.3. Heterogeneity of Urban Resource Endowments

The "resource curse" effect of resource-based cities has been confirmed by many scholars, so will the high-quality economic development effect of land transfer marketization also be plagued by the "resource curse"? In order to explore this problem, according to the division criteria of the State Council's National Sustainable Development Plan for Resource-based Cities (2013–2020), the sample cities are divided into resource-based cities and non-resource-based cities. Column (1) of Table 8 is the result of the return of urban resource endowment heterogeneity, which shows that the high-quality economic development effect of land transfer marketization is not obvious in resource-based cities, and the regression coefficient is not significant, while the effect is obvious in non-resource-based cities. The possible reason is that resource-based cities have deep-rooted problems of dependence and locking in the development path, and it is difficult to break through the shackles of the existing industrial structure, so the infiltration of land transfer marketization will be subject to increased resistance, and the high-quality economic development effect of land transfer marketization is weak. In contrast, land transfer in non-resource-based cities has a high degree of market-oriented acceptance, rapid development, and it is easy to realize the use of digital technology and promote the upgrading of industrial structure.

**Table 8.** Heterogeneity test regression results II.

| Variables | (1) | | (2) | | | |
|---|---|---|---|---|---|---|
| | Heterogeneity of Urban Resource Endowments | | Heterogeneity of Urban Features | | | |
| | | | The Level of Digital Financial Inclusion | | The Level of Fiscal Expenditure | |
| | Resource-Based Cities | Non-Resource-Based Cities | High | Low | High | Low |
| lnLT | −0.4433 | −0.3013 ** | −0.4146 ** | −0.5017 * | −0.5363 * | −0.5016 *** |
| | (0.0394) | (0.1969) | (0.2163) | (0.3063) | (0.3252) | (0.1135) |
| lnSLT | 1.0767 | 1.0643 ** | 1.0768 *** | 1.0267 * | 1.0742 * | 1.0637 *** |
| | (0.2395) | (0.2014) | (0.2738) | (0.1728) | (0.2816) | (0.2469) |
| Control Variables | control | control | control | control | control | control |
| R² | 0.3617 | 0.3568 | 0.3921 | 0.4112 | 0.4227 | 0.4053 |
| N | 2780 | 2780 | 2780 | 2780 | 2780 | 2780 |

Note: the brackets in the estimation results are the robust standard errors of individual clustering; *, **, ***, indicating a rejection of the original hypothesis at the significance levels of 10%, 5%, and 1%, respectively.

### 4.5.4. Heterogeneity of Urban Features

Adapting to local conditions is an important aspect that needs to be considered to efficiently play the market-oriented economic high-quality development effect of urban

land transfer. The marketization of land transfer and the high-quality development of urban economy are inseparable from urban financial investment and new infrastructure scale, this study is based on the "Digital Inclusive Finance Index" released by the Internet Finance Research Center of Peking University, indicating the level of urban inclusive finance, and the sample cities are divided into two equal parts: cities with high inclusive financial level and cities with low inclusive financial level. The proportion of urban fiscal expenditure in GDP is used to represent the level of urban fiscal expenditure, and the data are derived from the "China Urban Statistical Yearbook", which also divides the sample cities into two equal parts: cities with high fiscal expenditure levels and cities with low fiscal expenditure levels. Column (2) of Table 8 is the result of the return of the heterogeneity of the city's own characteristics, which shows that cities with a higher level of digital inclusive finance can play a more important role in the high-quality economic development of land transfer marketization, but compared with cities with high levels of urban fiscal expenditure, the high-quality economic development effect of land transfer marketization is more significant in cities with low fiscal expenditure. The possible explanation is that the high-level fiscal expenditure level of the city's own high-quality economic development means are diluted, and other ways dilute the role of land transfer market-oriented economic high-quality development, so land transfer marketization may play a greater role in the low fiscal expenditure level cities with a single economic high-quality development path.

## 5. Further Exploration: A Dual-Goal Constraint Perspective

According to the theoretical analysis mentioned above, the role of excessive economic growth goals and strict environmental targets on the marketization of land transfer on the high-quality development of urban economy is completely different, so this study explores the regulatory effect of dual goal constraints from an empirical perspective. Relying on the work report of China's prefecture-level municipal governments, the economic growth target constraint (EGC) of each local-level city is measured by the economic growth target value published in the government work reports in previous years, the interaction between the structure and the marketization of land transfer, and further distinguishing between the hard constraint and the soft constraint of the economic growth target [59], that is, when the economic growth target is formulated with the words "strive for", "above", "ensure", etc., this study is defined as the economic growth target hard constraint (HEGC), and when the interval description of "up and down" and "left and right" is adopted, it is defined as the Economic Growth Target Soft Constraint (SEGC). In terms of environmental target constraint measurement, this study measures the environmental target constraint (EOC) based on whether the energy consumption target is clearly put forward in the work report of the prefecture-level municipal government in previous years, and then constructs the interaction with the marketization of land transfer [60], and on this basis, the environmental target constraint is subdivided into direct environmental target constraints and indirect environmental target constraints, if the prefecture-level city meets the specific emission reduction target value specified in the government work report, and the assessment is announced in the next annual government work report. It is considered that the Direct Environmental Objective Constraint (DEGC) has been implemented, and in other cases, the Indirect Environmental Objective Constraint (IEOC). The data related to economic growth target constraints and environmental target constraints are derived from the original text of the "Government Work Report" of various cities in previous years and were obtained by hand.

Column (1) of Table 9 is the regression result under the economic growth target constraint, and the results show that the interaction coefficient between the economic growth target constraint and the land transfer marketization is significantly negative at the 10% level, indicating that the economic growth target constraint negatively regulates the impact of land transfer marketization on the high-quality development of the urban economy. Moreover, the inflection point value of land transfer marketization affecting the high-quality development of urban economy was increased from 0.2588 to 0.2921, indicating

that the economic growth target constraint delayed the inflection point of the impact of land transfer marketization on the high-quality development of urban economy from inhibition to promotion. From the perspective of soft and hard constraints on economic growth targets, it presents completely different results, and the negative adjustment effect of economic growth targets is obvious, which not only improves the inflection point value, but also causes the impact of land transfer marketization on the high-quality development of the urban economy to be delayed, while the soft constraints of economic growth targets have a positive adjustment effect and reduce the inflection point value, and the impact of land transfer marketization on the high-quality development of the urban economy is changed from inhibition to the inflection point of promotion. This shows that the excessively high economic growth target constraint weakens the high-quality economic development effect of the land transfer market, and the moderate economic growth target constraint is conducive to giving play to the high-quality economic development efficiency of the land transfer market.

**Table 9.** Dual-objective constraint regression results.

| Variables | (1) Regulation of Economic Growth Target Constraints | | | (2) Regulation of Environmental Target Constraints | | |
|---|---|---|---|---|---|---|
| lnLT | −0.8631 ** (0.3431) | −0.8372 ** (0.3463) | −0.4723 ** (0.3531) | −0.4511 ** (0.1513) | −0.5765 ** (0.2132) | −0.6323 ** (0.0386) |
| lnSLT | 1.4776 *** (0.5223) | 1.4621 * (0.5185) | 1.3801 ** (0.3418) | 1.1136 *** (0.1562) | 1.1824 ** (0.2014) | 1.3064 ** (0.3376) |
| LnLT * EGC | −0.0279 * (0.0136) | / | / | / | / | / |
| LnLT * HEGC | / | −0.0211 * (0.0073) | / | / | / | / |
| LnLT * SEGC | / | / | 0.0287 ** (0.0142) | / | / | / |
| LnLT * EOC | / | / | / | 0.0675 ** (0.0744) | / | / |
| LnLT * DEGC | / | / | / | / | 0.0452 *** (0.0581) | / |
| LnLT * IEOC | / | / | / | / | / | 0.0216 * (0.0113) |
| Control Variables | control | control | control | control | control | control |
| Year FE | YES | YES | YES | YES | YES | YES |
| City FE | YES | YES | YES | YES | YES | YES |
| $R^2$ | 0.4601 | 0.4213 | 0.4469 | 0.4373 | 0.4801 | 0.4712 |
| N | 2780 | 2780 | 2780 | 2780 | 2780 | 2780 |

Note: the brackets in the estimation results are the robust standard errors of individual clustering; *, **, ***, indicating a rejection of the original hypothesis at the significance levels of 10%, 5%, and 1%, respectively.

Column (2) of Table 9 is the regression result under the environmental target constraint, and the results show that the interaction coefficient between the environmental target constraint and the land transfer marketization is significantly positive at the 5% level, indicating that the environmental target constraint positively regulates the impact of land transfer marketization on the high-quality development of the urban economy. Moreover, the inflection point value of the marketization of land transfer affecting the high-quality development of the urban economy was reduced from 0.2588 to 0.2025, indicating that the environmental target constraints will shift the inflection point of the impact of land transfer marketization on the high-quality development of the urban economy from inhibition to promotion. From the comparison between the direct environmental target constraint and the indirect environmental target constraint, the positive adjustment effect of the direct environmental target constraint is significantly stronger. This shows that strict environmental target constraints strengthen the efficiency of high-quality economic development of land transfer marketization. At this point, Hypothesis 3 is fully verified.

## 6. Conclusions and Policy Implications

### 6.1. Conclusions

Based on the perspective of dual target constraints, the study analyzes the impact mechanism of land transfer marketization on the high-quality development of urban economy, uses the proportion of land bidding, auctioning, and listing transfer area to the total transfer area to measure the marketization level of land transfer in 278 cities above the prefecture level from 2011 to 2020, and measures the high-quality development of urban economy by reference to the regional development and people's livelihood index released by authoritative institutions. The static panel model and the SDM spatial econometric model study the nonlinear impact and spatial spillover effect of land transfer marketization on the high-quality development of urban economy and use the improved intermediary model and the moderating effect model to identify the mechanism between the two, as well as examine the regulatory effect of the dual constraints of economic growth goals and environmental objectives on the relationship between the two. The main conclusions are as follows:

(1) There is a significant U-shaped nonlinear relationship between the marketization of land transfer and the high-quality development of urban economy. From the static panel test, when the marketization level of land transfer is lower than 0.2588, the marketization of land transfer has an inhibitory effect on the high-quality development of the urban economy, and with the continuous improvement of the marketization level of land transfer, its inhibitory effect on the high-quality development of the urban economy gradually weakens. When the marketization level of land transfer crosses the inflection point value of 0.2588, its impact on the high-quality development of the urban economy changes from inhibition to promotion, and the promotion effect gradually increases with the improvement of the marketization level of land transfer. Judging from the SDM spatial effect test, the spatial spillover effect of land transfer marketization affecting the high-quality development of urban economy is small, mainly for the local area. This conclusion is still true after a series of tests for soundness.

(2) The resource allocation effect and the industrial structure upgrading effect are the main mechanisms of action. The mechanism test found that the contribution of the two major mechanisms to the marketization of land transfer on the high-quality development of the urban economy exceeded 50%, of which the contribution of the resource allocation effect was the largest, reaching 38.2166%, followed by the industrial structure upgrading effect of 20.5533%, but the existence of other mechanisms of action was still not excluded, such as technological innovation effect, energy efficiency improvement effect, and economic growth effect.

(3) The dual goal constraint plays a significant regulatory role in the relationship between the marketization of land transfer and the high-quality development of the urban economy. Moderate economic growth target constraints positively regulate the impact of land transfer marketization on the high-quality development of urban economy. Moreover, the inflection point value of land transfer marketization is advanced from 0.2588 to 0.1711, and the level of land transfer marketization is easier to reach the threshold of improving the high-quality development of urban economy, which strengthens the efficiency of high-quality economic development of land transfer marketization; excessive economic growth targets restrict the negative adjustment of the impact of land transfer marketization on the high-quality development of urban economy, and the inflection point value of land transfer marketization is extended from 0.2588 to 0.2921, weakening the efficiency of high-quality economic development of land transfer marketization.

In general, environmental target constraints positively regulate the impact of land transfer marketization on the high-quality development of urban economy. Moreover, the inflection point value of land transfer marketization is advanced from 0.2588 to 0.2025, and the level of land transfer marketization is easier to reach the threshold of improving the high-quality development of urban economy, which strengthens the efficiency of high-quality economic development of land transfer marketization; compared with indirect

environmental target constraints, the positive adjustment effect of direct environmental target constraints is significantly stronger, indicating that strict environmental target constraints are more conducive to strengthening the high-quality economic development effect of land transfer marketization.

(4) There are multiple heterogeneities in the impact of land transfer marketization on the high-quality development of urban economy. The market-oriented economic high-quality development effect of land transfer in cities north of the Qinling-Huaihe River is significantly higher than that of cities in the south. Compared with small- and medium-sized cities, large cities are more likely to promote the high-quality development of urban economy through land transfer marketization. Cities with high levels of financial inclusion and low levels of fiscal expenditure can significantly promote the high-quality economic development effect of land transfer marketization.

### 6.2. Policy Recommendations

The research conclusions of this paper provide an explanation based on the perspective of land resource allocation system for the "mystery of China's economic growth", enrich the relevant research on land resource allocation theory, and also provide a preliminary assessment of the economic growth performance of China's land system reform, which has important policy implications for deepening the reform of land use system.

(1) Deepen the market-oriented reform of land elements and give full play to the high-quality economic development effect of land transfer marketization. Urban land use should continue to adhere to the reform direction of the market-oriented land transfer system, enhance the transparency of land primary market transactions, standardize bidding, auction, and listing transfer, establish and improve the unified urban and rural construction land market, continuously expand the field of land transfer marketization, adjust and improve industrial land use policies, innovate use methods, promote the rational conversion of different types of industrial land, explore increasing the supply of mixed industrial land, and encourage high-quality projects to obtain land through market methods. Give full play to the basic role of market mechanisms in the allocation of land elements, realize the low-carbon transformation of urban economy, and promote the high-quality development of urban economy.

(2) Explore the combination of scientific and reasonable economic growth goals and environmental target constraints and give play to the positive role of dual goal constraints in the marketization of land transfer in the high-quality development of urban economy. When formulating annual targets, local governments should adopt soft constraints on economic growth targets as much as possible, avoid excessive pressure on economic growth targets, expand the scale of government debt financing, stimulate excessive investment, hinder the marketization of land transfer to play the efficiency of high-quality economic development, and inhibit the high-quality development of urban economies. At the same time, local governments should strengthen environmental target constraints, adopt strict environmental regulatory measures, give full play to the innovative supplementary effect and quality improvement effect of environmental target constraints, and effectively promote the synergy between land transfer marketization and environmental regulation. Specifically, on the basis of full research and technical simulation, the government should formulate a differentiated optimal combination of dual target constraints, achieve target heterogeneity management within the jurisdiction of the city, and give play to the guiding role of policy tools, rely on the difference between urban administrative level and urban scale, implement targeted special policies, expand the performance of market-oriented low-carbon governance of urban land transfer through policy dividends, and promote the high-quality development of urban economy.

(3) According to the objective facts that the characteristics of the city itself are quite different, scientifically release the high-quality economic development effect brought about by the market-oriented development of land transfer according to local conditions. There are differences in resource endowments, digital inclusive financial levels, and fiscal expen-

diture intensities among cities, resulting in cities having different degrees of high-quality development effects of the market-oriented economy of land transfer. Therefore, the government should adapt to local conditions, combined with the difference in whether the local area is a resource-based city, the level of inclusive financial development, and the intensity of fiscal expenditure. Formulate targeted policy plans to promote the effective integration of land transfer marketization and high-quality economic development, use digital technology to carry out full-life cycle digital transformation of traditional industries, enhance the adaptability of land transfer marketization and industrial structure, and improve the efficiency of land transfer marketization to promote the high-quality development of urban economy.

*6.3. Research Deficiencies and Prospects*

The research deficiencies in this paper are as follows: First, the "China Urban Statistical Yearbook" has been updated to 2021, but the statistical caliber of some data has not maintained continuity, so the high-quality development level of urban economy can only be evaluated from 2011 to 2020, and cannot reflect the latest results. Second, the research object of this paper is a prefecture-level city in China, and it has not been extended to the county level of China, so the richness and fullness of the study are missing. Third, when the mechanism is analyzed, this paper only finds two mechanisms of action, namely the resource allocation effect and the industrial structure upgrading effect, but the follow-up research may have more internal mechanisms. Finally, based on the purpose of the research and data considerations, this paper mainly considers the behavior of local governments at the city level, so there is not much attention paid to the enterprise level and construction of human settlement environment, the role of entrepreneurs as the driving force of the market economy, and the neglect of capital accumulation, which is the direction that future research needs to focus on and improve. In fact, a good living environment is also a means of high-quality development and high-quality urbanization. Building a beautiful living environment is not only to promote high-quality urbanization and promote high-quality urban development, but also the main means to promote high-quality national development, which is both the main content and an important means. Therefore, from the perspective of urbanization, building a better living environment is one of our important central tasks.

**Author Contributions:** Conceptualization, Z.Y. (Zhiqing Yan); methodology, Z.Y. (Zisheng Yang); software, Z.Y. (Zhiqing Yan); validation, Z.Y. (Zhiqing Yan); formal analysis, Z.Y. (Zhiqing Yan); investigation, Z.Y. (Zhiqing Yan); resources, Z.Y. (Zhiqing Yan); data curation, Z.Y. (Zhiqing Yan) and Z.Y. (Zisheng Yang); writing—original draft preparation, Z.Y. (Zhiqing Yan) and Z.Y. (Zisheng Yang); writing—review and editing, Z.Y. (Zhiqing Yan) and Z.Y. (Zisheng Yang); visualization, Z.Y. (Zhiqing Yan) and Z.Y. (Zisheng Yang); supervision, Z.Y. (Zhiqing Yan) and Z.Y. (Zisheng Yang); project administration, Z.Y. (Zhiqing Yan); funding acquisition, Z.Y. (Zisheng Yang). All authors have read and agreed to the published version of the manuscript.

**Funding:** This research was funded by the National Social Science Foundation of China (No.18VSJ023), Jiangxi University of Science and Technology School of Economics and Management Research Launch Project (JGBS202104).

**Institutional Review Board Statement:** Not applicable.

**Informed Consent Statement:** Not applicable.

**Data Availability Statement:** The data presented in this study are available on request from the corresponding author.

**Conflicts of Interest:** The authors declare no conflict of interest.

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
