# Peer review of "How the Marketization of Land Transfer under the Constraint of Dual Goals Affects the High-Quality Development of Urban Economy: Empirical Evidence from 278 Prefecture-Level Cities in China"

_sustainability, doi:10.3390/su142214707_

Round 1

Reviewer 1 Report

The article discusses land transfer under the constraint of dual goals affects the high-quality development of urban economy, using a marketing approach. Data analysis is based on multiple measurement models and multi-dimensional empirical analysis.

The authors concretely summarized and presented their findings. The methodology used also seems appropriate and well-argued. In the findings, they answered the research questions with arguments. 

The sample used and the longitudinal research show the coherence of the article.

In general, I believe that the article is prepared correctly and contains all the elements that a scientific contribution at this level needs.

Author Response

Point 1: The article discusses land transfer under the constraint of dual goals affects the high-quality development of urban economy, using a marketing approach. Data analysis is based on multiple measurement models and multi-dimensional empirical analysis.

The authors concretely summarized and presented their findings. The methodology used also seems appropriate and well-argued. In the findings, they answered the research questions with arguments. 

The sample used and the longitudinal research show the coherence of the article.

In general, I believe that the article is prepared correctly and contains all the elements that a scientific contribution at this level needs.

Response 1: First of all, thanks to the hard work of the external review experts, the valuable opinions put forward play an important role in improving the quality of the paper. Secondly, thank you very much for the recognition of the paper by the external review experts, we firmly believe that with your approval, this article will be more and more perfect.

Reviewer 2 Report

In general, the paper is very interesting especially regarding methodical approach in research on interdependence between market-oriented land transfer and high-quality economic development of urban areas (case study: China towns) - interesting as the separated things/issues.

Hypothesis are good, their verification OK., results presented... BUT... Is it possible to rely on these research results in city management and planning without proper consideration of environmental preconditions (environmental quality) for life quality of inhabitants in the city? Such environmental preconditions are not enough considered in these research or at least well discussed in final part of the paper.

Marketization of land transfer and / plus high stimulation of economic development can often make difficulties/threats  in planning the effective system of natural (green and water) areas in the city (as a counterweight to built-up areas) and creating of good / safe/ healthy conditions for residents. And what about with urban sustainable development and integration of its 3 components: economic, social and environmental ones? What about the concept of "city of tomorrow" (high all standards, high life quality for resisdents including high quality of environment (healthy and safe cities, concept of green city with high share of biologicaly active surface)?

The Authors of the paper should at least discuss well their research results regarding requirements in terms of environmental conditions wchich make human life possible in cities (especially in big cities, but not only in them).

Detailed notes:

Line 33: insert "." instead of ";".

U-shaped nonlinear relationship between marketization of land transfer and high-quality development of urban economy - is it possible to be shown on the chart or in different graphic way?

In general, the paper interesting and worth publishing, however under the condition of better discussion in the range of integration of land transfer and economic development with planning of rich natural system (system of green / water areas, biologically active surfaces) in the city - in terms of life comfort of residents.

Author Response

Point 1: In general, the paper is very interesting especially regarding methodical approach in research on interdependence between market-oriented land transfer and high-quality economic development of urban areas (case study: China towns) - interesting as the separated things/issues.

Hypothesis are good, their verification OK., results presented... BUT... Is it possible to rely on these research results in city management and planning without proper consideration of environmental preconditions (environmental quality) for life quality of inhabitants in the city? Such environmental preconditions are not enough considered in these research or at least well discussed in final part of the paper.

Marketization of land transfer and / plus high stimulation of economic development can often make difficulties/threats  in planning the effective system of natural (green and water) areas in the city (as a counterweight to built-up areas) and creating of good / safe/ healthy conditions for residents. And what about with urban sustainable development and integration of its 3 components: economic, social and environmental ones? What about the concept of "city of tomorrow" (high all standards, high life quality for resisdents including high quality of environment (healthy and safe cities, concept of green city with high share of biologicaly active surface)?

The Authors of the paper should at least discuss well their research results regarding requirements in terms of environmental conditions wchich make human life possible in cities (especially in big cities, but not only in them).

Response 1: Thanks to the effective opinions of external review experts, this paper mainly examines the impact of land transfer on the high-quality development of urban economy, and from the perspective of focus, this article involves more at the regional level and does not consider the resident level. However, we strongly agree with the recommendations of the external audit experts and have added a discussion of the quality of environmental quality at the level of urban dwellers in the final discussion.For example, we think a good living environment is also a means of high-quality development and high-quality urbanization. Building a beautiful living environment is not only to promote high-quality urbanization and promote high-quality urban development, but also the main means to promote high-quality national development, which is both the main content and an important means. Therefore, from the perspective of urbanization, building a better living environment is one of our important central tasks.

Point 2:Detailed notes:

Line 33: insert "." instead of ";".

U-shaped nonlinear relationship between marketization of land transfer and high-quality development of urban economy - is it possible to be shown on the chart or in different graphic way?

Response 2: Instead of has been added in line 33.

A simple U-shaped relationship diagram has been added to line 527.

Point 3:In general, the paper interesting and worth publishing, however under the condition of better discussion in the range of integration of land transfer and economic development with planning of rich natural system (system of green / water areas, biologically active surfaces) in the city - in terms of life comfort of residents.

Response 3: Thanks again to the external review experts, this part of the discussion has been increased.